# Injection 3D Concrete Printing (I3DCP): Basic Principles and Case Studies

**DOI:** 10.3390/ma13051093

**Published:** 2020-03-01

**Authors:** Norman Hack, Inka Dressler, Leon Brohmann, Stefan Gantner, Dirk Lowke, Harald Kloft

**Affiliations:** 1Institute of Structural Design, Technische Universität Braunschweig, Pockelsstr. 4, 38104 Braunschweig, Germany; leon.brohmann@tu-braunschweig.de (L.B.); s.gantner@hbk-bs.de (S.G.); h.kloft@tu-bs.de (H.K.); 2Institute of Building Materials, Concrete Construction and Fire Safety, Technische Universität Braunschweig, Beethovenstr. 52, 38106 Braunschweig, Germany; i.dressler@ibmb.tu-bs.de (I.D.); d.lowke@ibmb.tu-bs.de (D.L.)

**Keywords:** Injection 3D Concrete Printing, digital concrete, concrete 3D printing

## Abstract

Today, the majority of research in 3D concrete printing focuses on one of the three methods: firstly, material extrusion; secondly, particle-bed binding; and thirdly, material jetting. Common to all these technologies is that the material is applied in horizontal layers. In this paper, a novel 3D concrete printing technology is presented which challenges this principle: the so-called Injection 3D Concrete Printing (I3DCP) technology is based on the concept that a fluid material (M1) is robotically injected into a material (M2) with specific rheological properties, causing material M1 to maintain a stable position within material M2. Different to the layered deposition of horizontal strands, intricate concrete structures can be created through printing spatially free trajectories, that are unconstrained by gravitational forces during printing. In this paper, three versions of this method were investigated, described, and evaluated for their potential in construction: A) injecting a fine grain concrete into a non-hardening suspension; B) injecting a non-hardening suspension into a fine grain concrete; and C) injecting a fine grain concrete with specific properties into a fine grain concrete with different properties. In an interdisciplinary research approach, various material combinations were developed and validated through physical experiments. For each of the three versions, first architectural applications were developed and functional prototypes were fabricated. These initial results confirmed both the technological and economic feasibility of the I3DCP process, and demonstrate the potential to further expand the scope of this novel technology.

## 1. Introduction

While additive manufacturing (AM) is already a well-established technology and integral part of the production in many industrial sectors, the potential of AM to also become a key technology in the construction industry is clearly evident today. Compared to traditional concrete construction, AM offers a number of advantages: through the use of computer-controlled machines, building components can be mass-customized and produced individually in series, thus combining the benefits of traditional craftsmanship and industrialized production. In addition, digital control of 3D printing hardware allows the geometric complexity of components to be increased in order to improve component efficiency, for example by means of structural optimization. As such, material can be placed exclusively where it is structurally required. A further advantage of this technology is the complete elimination of conventional formwork, therefore minimizing the amount of work required for the formwork assembly, as well as the amount of construction waste caused by the disposal of the used formwork [1]. Current approaches of additive manufacturing in construction (AMC) are mainly based on the following methods: particle bed printing [2] and material extrusion [3]. A third emerging technology, the so-called Shotcrete 3D Printing, is based on material jetting [4]. 

Particle bed printing is a process where fine particles are selectively bound through the localized application of binder. For this, a thin layer of particles is evenly distributed inside a given building envelope. In a subsequent step a binder is deposited onto the particle layer, which selectively bonds the particles. Next, the particle bed is lowered, a new layer of particles is applied, and the binding process is repeated. In a layered manner, complex geometries can be fabricated, resting loosely inside the bed of unbound particles. This technology is used, for example, for the production of complex molds in metal foundry [5], but also in architecture as formwork for casting high resolution concrete parts [6]. Instead of using the printed element merely as formwork, there are also attempts to entirely print structural elements using large, room-sized particle bed printers [7]. 

In the extrusion-based approach to additive manufacturing, strands of fine concrete are printed layer-wise on top of each other, progressively creating a three-dimensional object. Here, predominantly two different concepts are pursued, firstly the contour crafting approach [8], and secondly the technique of 3D concrete printing [9]. Contour crafting is based on a lost formwork method where only the perimeter is printed, to create a hollow core element that is later filled with conventional concrete. Several research groups and industry professionals [10,11,12,13] have today adopted this method, also for the reason that it can be combined with the manual integration of reinforcement [14]. In the 3D concrete printing process, the entire concrete component is printed; however, channels for the placement of post-tensioning cables can be integrated in order to reinforce the component [15].

The third approach, Shotcrete 3D Printing, differs from extrusion-based concrete printing in the respect that the concrete is not deposited in strands, but is sprayed by means of compressed air [16]. This results in good layer adhesion, and the capability to spray around and hence embed structural reinforcement [17]. 

Each of these techniques features a distinct set of advantages, as well as process inherent limitations. 

Material extrusion, for example, offers high building rates. Depending on the printing system, state-of the-art concrete printers reach building speeds of up to 16 cm/s [18], corresponding to an extrusion volume flow of approximately 0.6 m^3^/h. Even higher volume flows of currently up to 1 m^3^/h can be achieved with the Shotcrete 3D Printing method. However, printing speed is inversely related to geometric resolution and surface quality. Accordingly, particle bed printing is, for example, the most geometrically versatile, but also the slowest additive manufacturing process.

A common feature of all three approaches is that the printed object is always built-up in horizontal layers. In this paper a novel 3D printing technology with concrete is introduced, which challenges the layered build-up and proposes a printing approach enabling more complex spatial printing trajectories. In the so-called Injection 3D Concrete Printing (I3DCP) method, one fluid material (M1) is robotically injected into another fluid material (M2) with similar rheological properties. In consequence, the extruded material M1 maintains in a stable position within material M2 without sinking or buoying. Accordingly, this process combines the advantages of the two techniques described above: firstly, the material M1 can be extruded at high building rates, and secondly, the surrounding liquid enables omnidirectional printing and thus comprehensive spatial freedom without restriction by gravity.

## 2. Basic Principles of Injection 3D Concrete Printing

In this paper the systematic investigation of three variations of Injection 3D Concrete Printing technique are described:3D printing of a fine grain concrete into a non-hardening suspension called “CiS” (Concrete in Suspension);3D printing a non-hardening suspension into a fine grain concrete called “SiC” (Suspension in Concrete);3D Printing a fine grain concrete with specific properties into another fine grain concrete with different properties, the so called “CiC” process (Concrete in Concrete).

With the first process version, CiS, a fine grain concrete is printed into a non-hardening suspension. As such, intricate truss structures and filigree concrete space frames can be fabricated, which are otherwise difficult or impossible to manufacture. Figure 1 shows the schematic fabrication method, as well as a possibly printed object.

The second process version SiC (suspension in concrete) is the reversal of the CiS principle: in the SiC process, a non-hardening suspension is printed into a vessel filled with fresh concrete. Subsequently, the suspension is removed, leaving cavities or channels. This can be used to functionally grade concrete components through differentiated density or to utilize them for physical element activation. Figure 2 shows schematically the SiC-fabrication process as well as a printed element. 

For the third version of this concept, concrete is used for both the extrusion and the supporting material, whereas the material properties differ according to functional needs (CiC, Figure 3). One possibility is to inject a high-performance concrete with high strength into a concrete with low strength. When injecting along the prevailing compression trajectories, a severely improved overall mechanical performance of the printed element is achievable. The special feature of the CiC method compared to the other variants is that both liquids form a permanent compound. 

## 3. State of the Art

The underlying concept, to continuously extrude material in spatial trajectories, was previously investigated with other materials, with and without a supporting medium. The Mataerial research project of the Institute for Advanced Architecture of Catalonia (IAAC) investigated the usage of a two-component thermosetting polymer in order to robotically print material in mid-air [19]. Concurrently, the Mesh Mould research at ETH Zurich investigated the use of thermoplastic materials, for example, Acrylnitril-Butadien-Styrol-Copolymere (ABS) and Polyactic Acid (PLA), to spatially print geometrically complex mesh structures [20]. In both techniques, the ability to print freely in air was limited by the hardening behavior of the material and the gravitational forces acting on the unsupported material cantilevering in the air. In 2014, a research team from Princeton University demonstrated a technique called “buoyant extrusion”, in which a thermoset material such as urethane plastic is printed in a container with sodium carbomer gel [21]. The materials can be adjusted so that the printed material slowly rises, sinks, or floats weightlessly in the other material. In the Large-Scale Rapid Liquid Printing project, researchers at the Massachusetts Institute of Technology are investigating the potential of this method for the large-scale industrial production of objects using high-quality industrial-grade rubbers, foams, and plastics [22].

These initial successes using polymers triggered testing of the concept with cement-bonded materials as well. For example, the French start-up Soliquid recently published a video of extruding a concrete in a container filled with gel [23]. Another team of researchers from Singapore University of Technology and Design (SUTD) robotically injected a reactive gas-forming powder, into a flat formwork filled with concrete, in order to create porous building elements [24]. These experiments concurred with the initial experiments at TU Braunschweig and the filing of the multi-material patent “Injection 3D Printing”, related to this paper [25] .

From the outset, the concrete-based I3DCP research at TU Braunschweig aimed to systematically investigate the method in its three conceptual variations—CiS, SiC, and CiC—in order to create a coherent system and to develop material combinations that make all three versions applicable for the construction industry. Here, a particular focus was directed to: the identification of process inherent applications in construction;the use and design of materials suitable for construction in terms of robustness, ecological, and economic factors;fundamental interactions between material behavior and fabrication process.

## 4. Experimental Setup

In the following sections, the fundamental and systematic exploration of the three versions of I3DCP is described. The methods were developed in the research seminar “Digital Building Fabrication Studio” (DBFS) at TU Braunschweig and were validated through a manifold of physical experiments. The following sections describe the robotic setup used, the digital workflows, and the materials developed. This is followed by a description of the experiments on each of the three variations of the I3DCP technique. 

### 4.1. Robotic Setup

All experiments were carried out on an UR 10 robot [26] using one of two custom-designed extruders. One of the extruders was based on a pneumatic material feed system, the other on a material feed system actuated by an electromechanically driven piston. Both end effectors featured specific advantages as well as limitations. The pneumatic extruder was robust and easy to handle, consisting of only a few components. The disadvantage, however, was that the material was fed indirectly by air pressure, and was therefore difficult to regulate precisely. In contrast, the electromechanic extruder allowed for defined control of the extrusion volume by positioning the piston with sub-millimeter precision. As a result, the extrusion process could be stopped and restarted at any time. This ability was additionally enhanced by the implementation of a pneumatically actuated pinch valve, mounted between the concrete hose and the nozzle. Both end effectors had a material capacity of 8 liters. Due to robot payload limitations, the material cartridge was decoupled from the robot and was stationary-mounted. An elongated interchangeable steel nozzle with a length of 80 cm and varying diameters from 8 to 16 mm was connected to the cartridge by a flexible rubber hose, and was attached to the robot via a 3D printed adapter. Both the electric and the pneumatic extruder were controlled via the analogue and digital outputs of the robot. The robotic setup is depicted in Figure 4.

### 4.2. Design and Control 

For design, path planning, simulation, and robot control an integrative design-to-fabrication approach was developed. For this purpose, the modeling software Rhino 3D [27] with its integrated programmable, graphical extension Grasshopper was used. Inside the Grasshopper environment, purpose-built plug-ins can be implemented or programmed for specific applications. Here, the “Robots” plugin [28], developed at the Bartlett School of Architecture, which contains a library of widely used robotic set-ups, was used to simulate and control the robot. Within this software environment, a continuous design-to-fabrication workflow was implemented in which the design geometry is automatically converted into a simulation and subsequently into robot instructions. 

### 4.3. Materials

There were three materials used in the present experiments: (a) fine grain concrete with an Ordinary Portland Cement (OPC, CEM I 52.5N) and a quartz sand with a maximum grain size of 2 mm; (b) ground limestone-sand suspension with a maximum grain size of 0.5 mm as non-hardening suspension; and (c) sand-gel suspension as non-hardening suspension with a maximum grain size of 0.5 mm. A detailed overview of all components used is given in Table 1, below. Table 2 summarizes the fabrication setup, parameters and materials used in the different experimental investigations.

## 5. Investigations, Results, and Discussion 

### 5.1. Concrete in Suspension (CiS)

Printing concrete in a suspension offers the potential to create lightweight, intricate spaceframe structures, with a high strength to weight ratio, which are difficult to manufacture in other ways. The focus of these investigations was directed firstly toward the design potentials of this approach, and secondly to develop material formulations that are technically and economically feasible, also for the harsh environment of the construction site. 

#### 5.1.1. CiS Calibration Process and Preliminary Studies

For calibration purposes a series of small-scale experiments was conducted. In order to allow a visual examination of the extrusion process, the fine grain concrete was printed into a transparent vessel (37.6 × 26 × 18.9 cm) filled with gel. The pneumatically actuated end effector was filled with 3 liters of fine grain concrete, and the air pressure was initially set to 2 bar. A simple, yet spatial, zig zag geometry was chosen for the initial experiment.

The printed structure was removed from the gel after 48 hours of curing. Already during this time, the following observations were made: only a few hours after printing, a white, dense, gel-skin developed around the concrete structure (Figure 5). At the same time, water accumulated on the surface of the gel. In this respect, it can be expected that the absolute position of the object in the gel changed, and that the gel deteriorated while in contact with the curing concrete. The decomposition of the gel restricts the reusability of the material, and approximately 5% losses can be expected after each printing process.

As the extrusion appeared very bulky and undefined, in the subsequent calibration steps, the air pressure was gradually reduced, while the robot speed was kept constant. This was repeated until the robot speed and extrusion rate resulted in a constant extrusion diameter corresponding to the 16 mm diameter of the nozzle. For this, an air pressure of 0.5 bar was found to be a suitable. Different space filling geometries were tested (Figure 6).

In order to prevent the decomposition of the gel, as well as the formation of the white skin, and to counteract progressive sinking of the printed structure, two alternative support mediums were developed and tested. Firstly, in an effort to achieve greater overall stability (in terms of the material decomposition as well as printing support), the same bulk volume of quartz sand was added to the pure gel. Additionally, a less sensitive and more cost-effective limestone-sand suspension was developed, which did not require the use of expensive gel at all. Both mediums were tested and compared in printing experiments (Figure 7).

For the gel-sand suspension no formation of a white skin was observed, and only a very small amount of water accumulated on the surface of the printing container. When extruding within the limestone-sand suspension, it was observed that furrows were created behind the printing nozzle, when moving through the gel. Those furrows saturated with concrete, leading to the creation of ridges above the print trajectory (Figure 7b). In further experiments this was corrected by increasing the water content by 3%. In contrast to the gel-based suspensions, no problems with water going through the limestone-sand suspension on the suspension’s surface were observed at all. 

#### 5.1.2. CiS Demonstrator

To validate the CiS principle, a medium-sized object was developed for production in a 50 × 50 × 50 cm container. For this, a formwork was filled with approximately 110 liters of the non-hardening limestone-sand suspension. The object represents a 14-sided polyhedral spaceframe structure derived from a hexagonal base (Figure 8a). 

The printing path and sequence, as well as the robot simulation, are depicted in Figure 8b,c. The total path length was calculated to be 13.4 meters, resulting in an estimated printing time of 2:41 minutes, printed with a robot speed of 5 m/min. The printing process is schematically shown in Figure 9. 

#### 5.1.3. CiS Results 

After three days the object was removed from the limestone-sand suspension (Figure 10a). The vast majority of nodes were connected monolithically (24 out of 27), and the overall structure was stable. A few nodes however, especially on one side of the upper edge, did not connect well, leaving small gaps between the spaceframe elements. One node was entirely disconnected from the edge above, suggesting that the position of the material changed or was actively displaced during printing (Figure 10c). In addition, a slight expression of ridges above the extrusion was observed in this experiment (Figure 10b). This indicates that also in this experiment the supporting suspension was too stiff, and concrete flowed into the furrows that the nozzle had created while moving. Hence, the yield stress and viscosity of the suspension needs to be decreased, while a higher viscosity and yield stress of the injected fine grain mortar would reduce the appearance of the ridges. 

### 5.2. Investigations on Suspension in Concrete (SiC)

The potential of the SiC approach is to create concrete elements which are graded in porosity, or which can integrate geometrically complex systems of channels or ducts. These features could, for example, be used for aspects of structural performance, material savings, technical installations, or for component activation and building physics. In this approach the voids are created by printing a non-hardening suspension acting as displacement bodies directly into the fresh concrete. The challenges here were firstly, calibrating the material properties in such a way that the material remained stable within the concrete and generated well-defined voids, and secondly, designing the channel layout in a manner that the displacement material could drain off after the concrete had cured. 

#### 5.2.1. SiC Calibration Process and Preliminary Studies

A preliminary setup was developed for material and process calibration that allowed immediate visual feedback. For this purpose, a 30 × 30 × 4 cm³ formwork was built, in which the 30 × 30 cm² sides were closed with Plexiglas. For an initial test a centrically placed regular 6 × 6-point grid was generated, and at each point gel was injected into the fine grain concrete for a certain amount of time. The electromechanic piston extruder was used, due to the advantage that the material volume can be controlled more precisely. Compared to the pneumatic extruder, the material flow could be stopped and started at any time, whereas the pneumatic extruder continually builds up or releases pressure. This is particularly important if only material is to be injected selectively, for example as discretely enclosed volumes or voids. However, the initial experiment showed that simply stopping the piston does not automatically stop the material from flowing. Therefore, as a side-effect, channels were created when moving the nozzle from one point to the next.

Moreover, it was recognized that the gel did not have the capacity (density) to displace the concrete evenly (Figure 11b) and that the same white skin as in the CiS approach appeared inside the voids (Figure 11c). As a consequence of these findings, firstly the pneumatic pinch valve for abruptly stopping the material flow was implemented, and secondly the alternative mediums (sand-gel suspension and limestone-sand suspension) were subsequently tested. During these tests it was observed that the limestone-sand suspension was not easily extrudable and segregated when pressure was exerted. Hence, for the following experiments a gel-sand mix was tested and a ratio of 1:0.96 was found to be ideal. 

#### 5.2.2. SiC Demonstrator 

In order to demonstrate the unique potentials of the SiC process, a series of demonstrators were developed and realized. The design of a perforated facade panel was selected as the architectural case study. For this, a larger format of 50 × 50 cm^2^ was chosen, whereas the thickness of the panel was maintained. In contrast to the preliminary tests, in this series a differentiation of the size of the displacement bodies was deliberately made and realized by varying the duration of the extrusion at each point. Moreover, the pattern was changed from a regular 6 × 6 configuration to a diagonal pattern, where one row of points was shifted horizontally by half the distance between two points (Figure 12a,b). The point pattern was computationally generated, and automatically sorted from bottom to top, so that it could be printed without self-intersections. For each of the demonstration panels the difference between the total volume of the displacement bodies and the formwork capacity was calculated. The corresponding quantity of concrete was then prepared and poured into the formwork before printing (Figure 12c).

#### 5.2.3. SiC Results 

By using the gel-sand mixture instead of the pure gel for displacing the concrete, significantly better results were achieved. The concrete was displaced evenly, creating almost spherical voids. Slight geometrical distortions were possibly due to the fact that the suspension was not printed freely into the concrete, but against the plexiglass panel. In order to verify this assumption, a volumetric sample would have to be produced and then cut open. In addition to the voids, some panels also had an integrated channel system that connected the voids. In this specific demonstrator those channels were intended to be used as an integrated watering system for façade greening elements of physical component activation (Figure 13b).

### 5.3. Investigations Concrete in Concrete (CiC)

The potential of the CiC method is the ability to locally strengthen a concrete building element by injecting a high strength concrete into a lower grade material, for example, recycled concrete. In this process the high-strength concrete is injected along the compression trajectories within a concrete component. This local reinforcement makes it possible to construct more efficient structures, as the internal structure can be topologically optimized and expensive high-strength material can be used, only where it delivers maximum effect.

#### 5.3.1. CiC Calibration Process and Preliminary Studies

The aim of this experiment was to visualize the local differentiation of material properties, by printing a batch of pigmented concrete into a batch of the same, non-pigmented concrete. Here, 6 g of black pigment was added to every kg of dry premixed fine grain concrete. Regarding the appropriate concrete rheological properties for both batches, it was possible to build on the experience already gained in the CiS and SiC process. Accordingly, for extruding the pigmented concrete, the pneumatic piston extruder was used. 

#### 5.3.2. CiC Demonstrator

As proof of concept, a rectangular concrete beam measuring 80 × 15 × 20 cm^3^ was topologically optimized for a four-point bending scenario. Based on the resulting geometry, a continuous path was manually created (see Figure 14a). The total path length was measured to be 3.2 meters. With a robot speed of 5 m/min and a printing time of 39 seconds, a material consumption of 0.64 liters for printing was calculated. Accordingly, the formwork was filled with 23 liters of fine grain concrete, causing the formwork to be almost filled completely after the pigmented concrete was injected. Prior to pouring, two 10 mm reinforcement bars were placed in the tension zone of the beam, approximately 3 cm away from the outer surface. In order to avoid collisions with the reinforcement, the printing path was placed centrally within the formwork (Figure 14b). 

#### 5.3.3. CiC Results

The printing process for this demonstrator took merely 39 seconds. The component was demoulded after 48 hours of curing and was subsequently cut in transverse and longitudinal directions. The cuts show that the printed geometry was clearly defined and positioned correctly within the cast element (Figure 15). 

## 6. Potentials and Challenges of I3DCP 

All experiments described above represent a proof of concept and are a starting point for further research into Injection 3D Concrete Printing. It is believed that each of the process variations has a unique potential for the application in construction. This is particularly true as these initial tests showed that the process is fast and that the material combinations developed in this research are inexpensive and economically well within the range of common building materials. For example, by substituting the pure gel suspension (3.7 €/kg) with a limestone-sand suspension (<0.08 €/kg), the costs were reduced by almost two orders of magnitude. Moreover, all the methods presented in this paper are based on standard concrete technologies, for example, standard pumps and formwork systems. Therefore, all processes are considered scalable and, more importantly, economically feasible on the large scale of construction. In addition, the dimensional limitations are comparable to current precast concrete production or in-situ concrete constructions.

The unique potentials of each of the process variation and specific applications in construction are outlined in the following paragraphs. The challenges associated with up-scaling must also be identified and addressed in future research. 

### 6.1. CiS, Potentials and Challenges 

The CiS process offers the potential to radically expand the design space of concrete constructions. With contemporary techniques, be it conventionally casted or 3D printed, concrete elements are predominantly conceived as flat or curved surface structures (with the exception of columns). The CiS process, however, makes it possible to dissolve concrete structures into highly differentiated intricate spatial structures. This can be of practical use when, for example, structural performance is the main requirement, and other features, like, for example, the enclosure of a thermal envelope, are of secondary importance. In such cases the structure can be reduced to solely loadbearing members, and locally differentiated spaceframe structures can be printed with a minimum of material used. Moreover, and in addition to the potential of globally differentiating the spatial structures, the process enables the local differentiation of each individual truss member by altering the robot speed and extrusion rate during printing. First experiments have verified this approach, and are depicted in Figure 16a. 

Nevertheless, the structural performance of the systems depends not only on mere geometry but also on other parameters, like node connectivity, and on the integration of reinforcement. With the former, it must be ensured in each node that the materials bind monolithically to each other and that no separating layer is formed, as occasionally observed in the experiments. Regarding the integration of reinforcement, one major advantage here compared to the layered build-up of conventionally 3D printed structures, is the possibility to extrude material in line with the prevailing force trajectories. This unique feature makes it possible to co-extrude reinforcement, such as a continuous fiber or steel cables within the concrete strand. Initial tests have been carried out and are currently being developed further (Figure 16b).

Finally, a future challenge is to investigate the process on a larger scale. In this respect, two scenarios are considered possible. Firstly, to scale up the entire system and to print in very large containers filled with suspension, and secondly, to modularize the structures and build large constructions from smaller components. As the CiS process is believed to be most efficient in prefabrication, a modularization based on the maximum transportation size seems to be the most feasible fabrication scenario. Nevertheless, particularly as the limestone-sand suspension is very robust (compared to gel), this process would be feasible for onsite production as well. In that case larger spaceframes would be printed in an onsite factory, and be lifted into place using cranes. 

### 6.2. SiC, Potentials and Challenges 

The two main features which were explored in the SiC process were, firstly, the gradation of concrete by locally injecting discrete displacement bodies, and secondly, the integration of continuous channels within a solid building element. Both features offer a vast potential for practical application in construction. In addition to the possibility of making buildings generally lighter, the production of lighter prefabricated parts also offers considerable potential for saving energy during transportation to the building site. Besides saving weight, and hence resources, SiC structures also facilitate the integration of additional component functionalities. For example, through the integration of channels and ducts, active cooling or heating of building elements can be facilitated. Moreover, integrated channels may be used for structural purposes, for example, for the integration of post-tensioning cables. 

With regard to the location of production, both prefabrication and in situ fabrication seem possible. In further development, however, research will concentrate on prefabrication. This is due to the fact that the release and reuse of the suspension from the hardened concrete can be better controlled in a factory setting. 

### 6.3. CiC, Potentials and Challenges 

The CiC principle fosters the fabrication of high-performance components using mostly lower grade materials. For example, a normal concrete can be locally reinforced by injecting high-performance concrete in exceedingly stressed areas. This shows potential for a variety of practical applications. For example, the load paths in a component could be traced using a higher-quality material, so that the load-bearing capacity of the component increases along these paths. In the case of walls, for example, these would be areas in the immediate vicinity of doors or window openings. In addition, this method could also be used to locally reinforce areas of direct load introduction, for example in the supports where beams rest on a wall. 

While the CiC experiments described in this paper essentially used the same concrete, merely with the addition of black pigments, the challenge in further investigations will be the use of concrete with vastly differing mechanical properties. The challenge here will be maintaining their printing and bonding compatibility. In addition to using lightweight concrete as a support medium, the use of recycled concrete will also be subject to future investigation. 

In terms of structural design, currently predominantly continuous geometries have been printed into the formwork. This approach could be extended towards printing strategies in which the properties of a building element change gradually from property A to property B. For this, an injection strategy similar to the one described in the SiC approach will be investigated.

Of all three process variations, the CiC process seems to be most likely to be implemented in situ, for example, by mounting a lightweight robot onto standard concrete formwork, in order to locally inject high performance mortar into cast-in place concrete constructions. However, here too there is great potential for prefabrication, especially in view of the fact that this process can be combined and hybridized with the SiC process. 

## 7. Summary and Outlook

In this paper, an overall concept for a novel, so-called Injection 3D Concrete Printing technology was presented, which is based on the injection of a fluid material into another fluid material with specific rheological properties. Three variations of this method were described and evaluated: A) Concrete in Suspension (CiS): injecting a fine grain concrete into a non-hardening suspension; B) Suspension in Concrete (SiC): injecting a non-hardening suspension into a fine grain concrete; and C) Concrete in Concrete (CiC): injecting a fine grain concrete with specific properties into a fine grain concrete with different properties. Each of these variations of the Injection 3D Concrete Printing concept was verified through numerous pre-studies as well as a final demonstration object for each approach. Within this framework a particular focus was directed to the scalability and economic feasibility of the 3D printing processes. For this, material combinations were developed which meet construction requirements in terms of economy and robustness. 

While each of the three variants is associated with specific potentials and challenges, there are characteristics in common between the different processes that need to be investigated in the further course of research. Most importantly, the fabrication process needs to be scaled from lab environment to real scale. In particular, the extruder technology needs to be upgraded to meet reliable and robust industrial standards, and to be able to extrude larger quantities of concrete without interruptions. Moreover, the digital geometry generation and path planning should be extended to include a structural analysis of the generated geometry, as well as an automated feasibility check of the robotic trajectory, in order to avoid intersections of the already printed geometries. The rheological properties (yield stress and viscosity) of the materials need to be specified to enable geometric precision of the printed objects. In addition, the amount of creep and shrinkage of the extruded material has to be investigated in future research. 

Another aspect for further investigation is precise control over the increasing volume during the printing progress: the more material is injected into the other, the more the volume increases. The mechanisms of the rising material level and the associated positional changes of the already injected material will have to be investigated more closely in future research. 

Following these initial proof of concept studies, in future research the structural properties will have to be quantified. For all versions of the I3DCP process, specific benchmarking, e.g., regarding the weight/performance ratio, must be developed and performed. The implementation of the process in the construction industry can only be realized in the medium term through quantified comparison.

Finally, it should be stressed that the Injection 3D Concrete Printing method depends on a close interaction between the fabrication process and material parameters, and that these aspects cannot be considered separately. Hence, a close collaboration of material science, structural design, and process engineering is required to successfully advance the research. 

## 8. Patents

A patent is filed under: N. Hack, D. Lowke, H. Kloft, Injection 3D Printing, DE 10 2019 105 596.2, 2019. https://www.ezn.de/ezn-patent/injection-3d-printing/.

## Figures and Tables

**Figure 1 materials-13-01093-f001:**
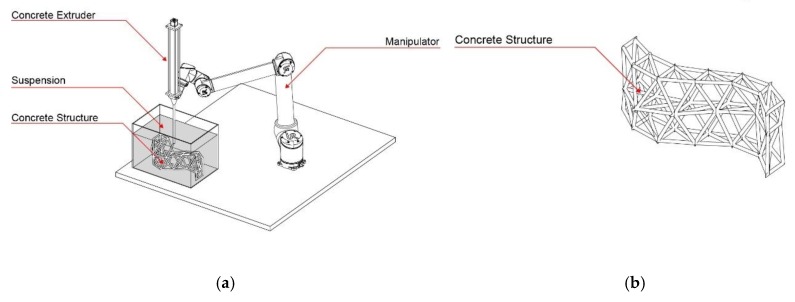
Concrete in Suspension (CiS): (**a**) fabrication process diagram; (**b**) printed lattice structure.

**Figure 2 materials-13-01093-f002:**
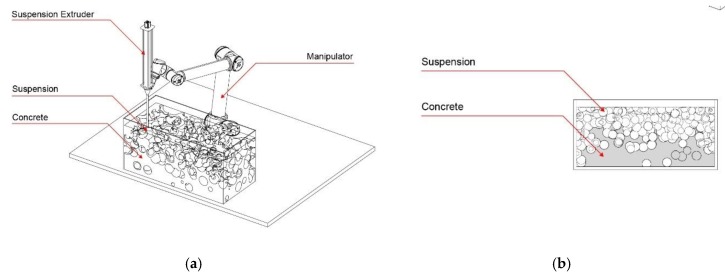
Suspension in Concrete (SiC): (**a**) fabrication process diagram; (**b**) printed graded element.

**Figure 3 materials-13-01093-f003:**
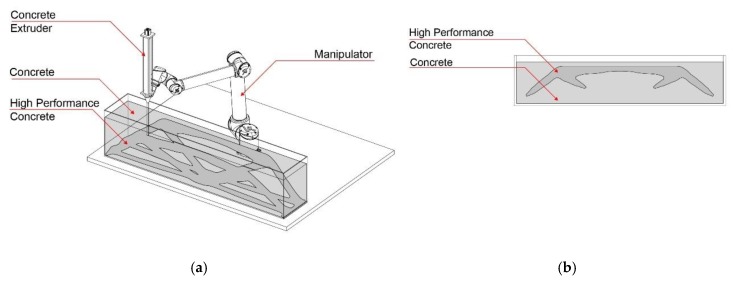
Concrete in Concrete (CiC): (**a**) fabrication process diagram; (**b**) section through a locally strengthened element.

**Figure 4 materials-13-01093-f004:**
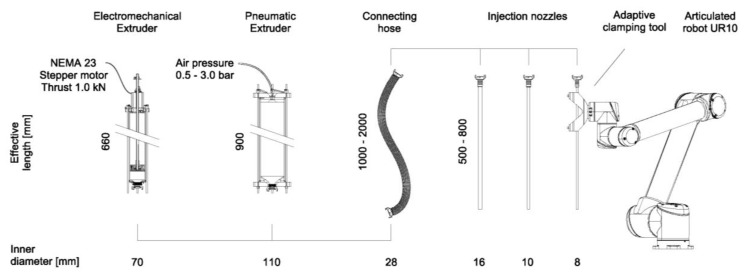
Robotic setup: electromechanical and pneumatic extruder, hose, nozzles with different diameters, and robot with nozzle.

**Figure 5 materials-13-01093-f005:**
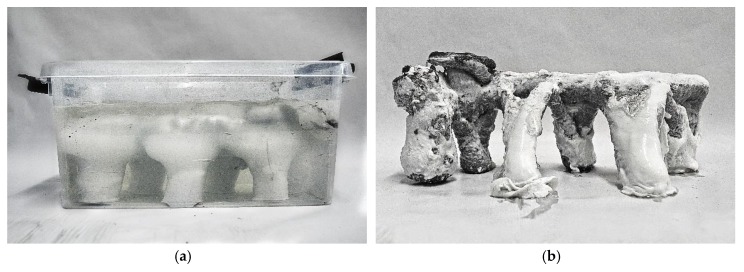
Initial experiment for the purpose of calibration: (**a**) zig zag geometry printed into gel, (**b**) the object after 48 hours of curing inside the gel.

**Figure 6 materials-13-01093-f006:**
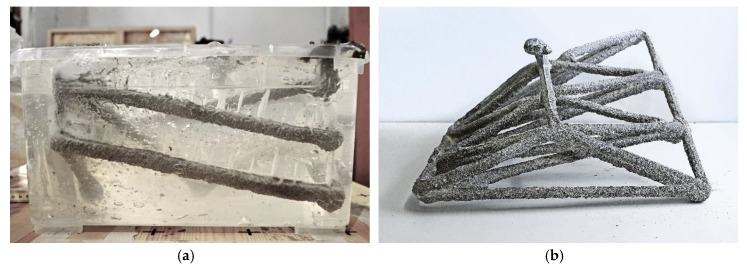
Results of the calibration process: (**a**) a spiraling geometry printed in gel; (**b**) a more complex geometry printed in a 1:1 gel-sand suspension.

**Figure 7 materials-13-01093-f007:**
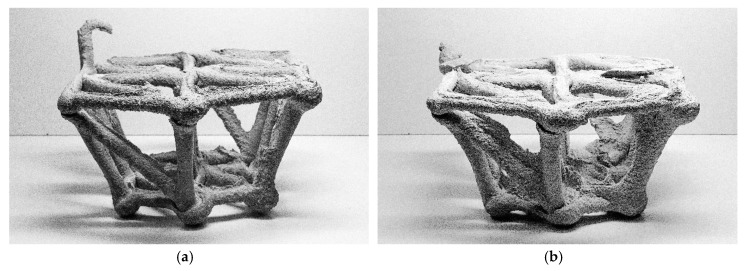
Objects printed in two different support mediums: (**a**) printed in a gel-sand suspension; (**b**) printed in a limestone powder quartz sand suspension.

**Figure 8 materials-13-01093-f008:**
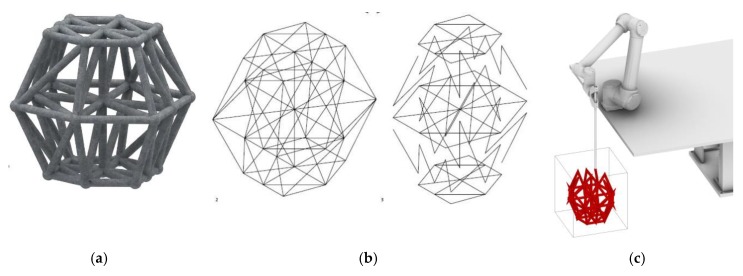
Object for process verification: (**a**) rendered structure, (**b**) wireframe model of the printing path, (**c**) robot simulation.

**Figure 9 materials-13-01093-f009:**
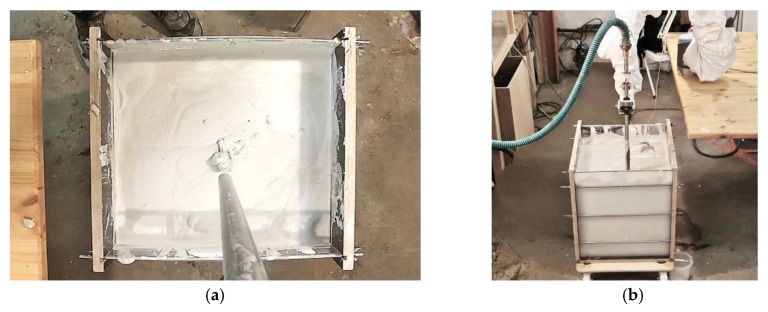
Printing process in the limestone-sand suspension: (**a**) closeup; (**b**) general setup with formwork placed on the ground.

**Figure 10 materials-13-01093-f010:**
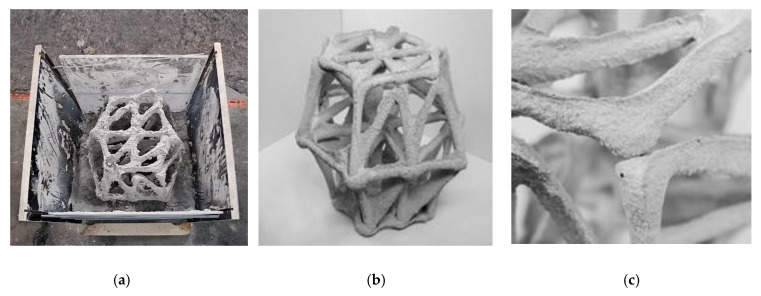
CiS demonstrator: (**a**) stripping the object from the formwork; (**b**) final object; (**c**) detailed view of a knot with a deficient connection.

**Figure 11 materials-13-01093-f011:**
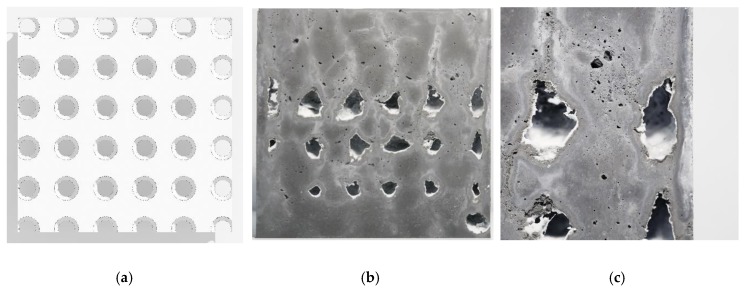
Initial experiment for calibration: (**a**) 6 × 6-point pattern; (**b**) result of the injection process; (**c**) closeup of the irregular voids with white skin and the unintended channels above.

**Figure 12 materials-13-01093-f012:**
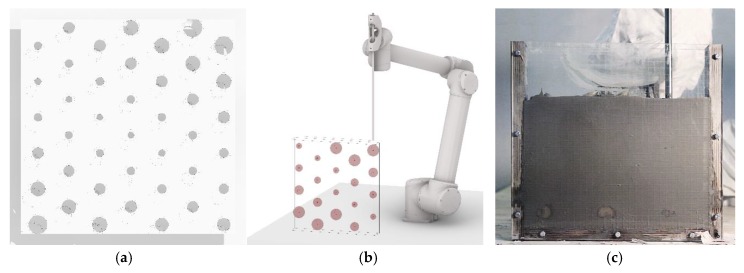
Object for process verification, SiC: (**a**) shifted pattern; (**b**) path panning and simulation; (**c**) fabrication process injecting gel-sand into a fine grain concrete.

**Figure 13 materials-13-01093-f013:**
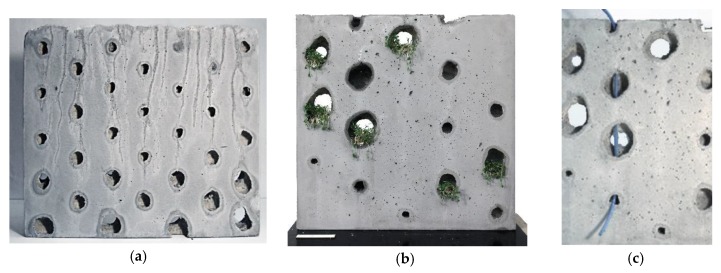
SiC demonstrators: (**a**) panel with displacement bodies gradually decreasing in size towards the top; (**b**) concept for façade planting; (**c**) integration of continuous channels connecting the voids for watering.

**Figure 14 materials-13-01093-f014:**
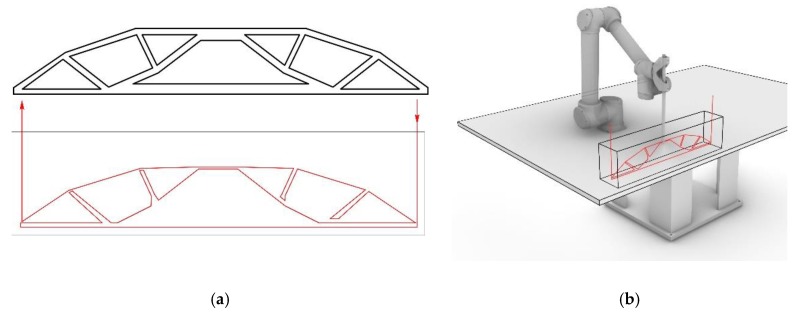
Object for process verification, (CiC): (**a**) topology optimized beam (top), robotic fabrication path (bottom); (**b**) simulation of the fabrication process.

**Figure 15 materials-13-01093-f015:**
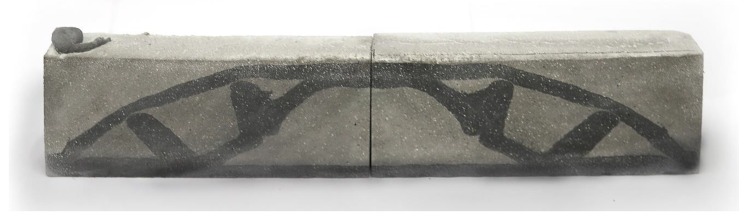
Longitudinal section through the CiC sample.

**Figure 16 materials-13-01093-f016:**
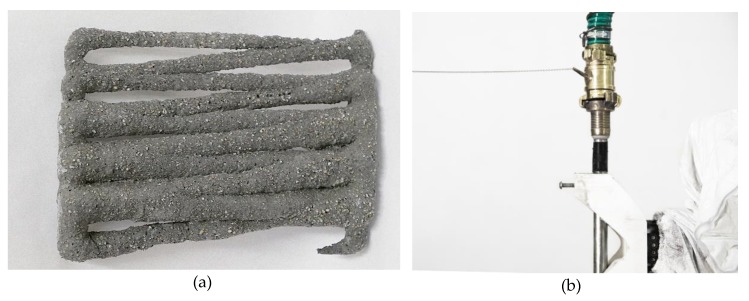
Further investigations: (**a**) differentiated extrusion thickness; (**b**) initial experiments testing the co-extrusion of reinforcement.

**Table 1 materials-13-01093-t001:** Mixture compositions of used material.

Components	Fine Grain Concrete	Limestone-Sand Suspension	Sand-Gel Suspension
	(kg/m³)	(kg/m³)	(kg/m³)
Portland Cement (CEM I 52.5N)	450.8	-	-
Ground limestone (LS)	-	695.6	-
Puzzolan	144.3	-	-
Silica fume	27.1	-	-
Aggregate, d = 0–0.5 mm	-	1237.0	727.1
Aggregate, d = 0–2 mm	1064.0	-	-
Water	342.8	275.6	-
Ultra Sonic gel	-	-	727.1
PP-Fibres	2.7	-	-
Additives (solid)	29.8	-	-
Superplasticizer (liquid)	-	1.0	-
Pigment (Iron Oxide, Black) ^1^	10.08	-	-

^1^: Used only for the extruded material in the CiC process

**Table 2 materials-13-01093-t002:** Overview of the fabrication setup, parameters and materials used during the different investigations.

^1^ Preliminary Studies ^2^ Demonstrator	Extruder	Robot Speed	Extrusion Volume	Injection Material	Support Material
**CiS**	pneumatic	5 m/min	1 l/min (0.5 bar)	Fine grain concrete	^1^ Gel and sand-gel suspension ^2^ Limestone-sand-suspension
**SiC**	electro-mechanic	5 m/min	1 l/min	Gel-sand suspension	Fine grain concrete
**CiC**	pneumatic	5 m/min	1 l/min (0.5 bar)	Pigmented fine grain concrete	Fine grain concrete

^1^ Material used during the exploratory phase of the process. ^2^ Material used for the fabrication of the final demonstrators.

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
