# Peer review of "Injection 3D Concrete Printing (I3DCP): Basic Principles and Case Studies"

_materials, 2020, doi:10.3390/ma13051093_

Round 1

Reviewer 1 Report

It is very interesting paper describing a noble injection methodology. Especially, it seems that the CiC process can have a large potential with further development of related technologies. The manuscripts have been read thoroughly but it seems that it is ready for a potential publication. 

Author Response

Thank you for your comments, your confidence in the quality of the paper pleases us very much.

Reviewer 2 Report

This paper presents a new and novel approach to enable 3D printing of concrete. The paper seems to be designed to showcase a few prototypes and then layout some of the observations and challenges observed during the fabrication of such prototypes. While the paper does not provide full details or findings, the paper is timely and of merit. The following minor items are to be addressed before the manuscript can be published:

  • Line 39, add a reference number.
  • The introduction is very short and lacks details on previous works as well as on the application of 3D printing. The author may strengthen this section with a discussion on few recent works in 3D printing (i.e. modular construction, space construction etc.). Some of these can be found in the following articles and the authors may option to use them or use others if deemed irrelevant.
    • https://doi.org/10.3390/jcs3030088
    • https://doi.org/10.3390/ma12060902
    • https://doi.org/10.1016/j.pmatsci.2019.100577

  • Add a few examples to how CiS, SiB, CiC are used in real applications. For example, will CiS work to print concrete under water?
  • Given that the authors have patented their work. If possible, comment on observations from material testing i.e. compressive strength as obtained using the three defined methods, workability and cohesion etc.
  • Please add a few lines on costs associated with printing such concrete using the above three methods.
  • How easy is to apply these methods in situ? How common are the raws needed for such printing?

Reviewer 3 Report

Dear authors, thank you for the interesting paper focused on 3D printing of concrete structures - at that time on a small scale yet. I think that it can be partially a future of concrete structures making.

I have the comments:

Are presented methods usable in practice? In all cases, it is needed to prepare a big volume of receiver for printing - with suspension (gel) or concrete. What is the maximum recommended dimension of the member done by these methods?

Method CIS - it is needed a lot of suspensions - are any problems with water going through suspension on suspension surface?

What about the shrinkage and creep characteristics of your concrete? Was it investigated?
